# Prognostic value of OHCA, C-GRApH and CAHP scores with initial neurologic examinations to predict neurologic outcomes in cardiac arrest patients treated with targeted temperature management

**Hyun Soo Kim[1]☯, Kyu Nam Park[1], Soo Hyun Kim[2], Byung Kook Lee[3], Sang Hoon Oh[1], Kyung Woon Jeung[3], Seung Pill Choi[2], Chun Song Youn[1]***

**1** Department of Emergency Medicine, Seoul St. Mary Hospital, College of Medicine, The Catholic University of Korea, Seoul, South Korea, **2** Department of Emergency Medicine, Eunpyeong St. Mary Hospital, College of Medicine, The Catholic University of Korea, Seoul, South Korea, **3** Department of Emergency Medicine, Chonnam National University Medical School, Gwangju, Korea

☯ These authors contributed equally to this work.
* ycs1005@catholic.ac.kr

## Abstract

### Objective

The aim of this study in out-of-hospital cardiac arrest (OHCA) patients treated with targeted temperature management (TTM) was to evaluate the prognostic value of OHCA, C-GRApH, and CAHP scores with initial neurologic examinations for predicting neurologic outcomes.

### Methods

This retrospective study included OHCA patients treated with TTM from 2009 to 2017. We calculated three cardiac arrest (CA)-specific risk scores (OHCA, C-GRApH, and CAHP) at the time of admission. The initial neurologic examination included an evaluation of the Full Outline of UnResponsiveness brainstem reflexes (FOUR_B) and Glasgow Coma Scale motor (GCS_M) scores. The primary outcome was the neurologic outcome at hospital discharge.

### Results

Of 311 subjects, 99 (31.8%) had a good neurologic outcome at hospital discharge. The OHCA score had an area under the receiver operating characteristic curve (AUROC) of 0.844 (95% confidence interval (CI): 0.798–0.884), the C-GRApH score had an AUROC of 0.779 (95% CI: 0.728–0.824), and the CAHP score had an AUROC of 0.872 (95% CI: 0.830–0.907). The addition of the FOUR_B or GCS_M score to the OHCA score improved the prediction of poor neurologic outcome (with FOUR_B: AUROC = 0.899, p = 0.001; with

**Data Availability Statement:** All relevant data are within the manuscript and its Supporting Information files.

**Funding:** This research was supported by Basic Science Research Program through the National Research Foundation of Korea (NRF) funded by the Ministry of Education (2018R1D1A1B07047594), awarded to CSY.

**Competing interests:** The authors have declared that no competing interests exist.

**Abbreviations:** AUC, area under the curve; AUROC, area under the receiver operating characteristic curve; BLS, basic life support; CA, cardiac arrest; CI, confidence interval; CPC, cerebral performance categories; CPR, cardiopulmonary resuscitation; CT, computed tomography; FOUR_B, Full Outline of UnResponsiveness brainstem reflexes score; GCS_M, Glasgow Coma Scale motor score; ICU, intensive care unit; OHCA, out-of-hospital cardiac arrest; ROC, receiving operating characteristic; ROSC, return of spontaneous circulation; TTM, targeted temperature management; WLST, withdrawal of life-sustaining therapy.

GCS_M: AUROC = 0.880, p = 0.004). The results were similar with the C-GRApH and CAHP scores in predicting poor neurologic outcome.

## Conclusions

This study confirms the good prognostic performance of CA-specific scores to predict neurologic outcomes in OHCA patients treated with TTM. By adding new variables associated with the initial neurologic examinations, the prognoses of neurologic outcomes improved compared to the existing scoring models.

## Introduction

Globally, out-of-hospital cardiac arrest (OHCA) is a major public health problem that affects between 250,000 and 300,000 patients annually, with an average incidence of 55 OHCAs per 100,000 adults and a low (7%) survival rate [1]. Due to advances in post-cardiac arrest (CA) management during the past decades, therapeutic hypothermia, called targeted temperature management (TTM), is the standard treatment of choice for CA patients, demonstrating an improvement in mortality and neurologic outcomes [2, 3]. However, despite intensive medical treatments, due to hypoxic damage to the brain and ischemia-reperfusion phenomena leading to a systemic inflammatory response, approximately two-thirds of initially resuscitated patients subsequently die in the hospital [4].

Accurate prediction after cardiac arrest is essentially needed to minimize the risk of erroneous prognostication of poor outcome. Withdrawal of life-sustaining therapy (WLST) after return of spontaneous circulation (ROSC) in Korea is legally prohibited, and early outcome prediction is crucial in regard to the proper distribution of limited medical assets. In other words, it is possible to allocate critical care resources by predicting the patient prognosis early.

To date, various scoring models have been suggested for predicting the neurologic outcomes after CA [5–7]. In the initial development of estimation models, an OHCA score was suggested in France [5]. The model included each parameter that was available upon admission to the intensive care unit (ICU), including initial rhythm, estimated no-flow (time from collapse to the initiation of cardiopulmonary resuscitation (CPR)) and low-flow intervals (time from CPR to the return of spontaneous circulation (ROSC)), blood lactate levels, and creatinine levels. Another model is the C-GRApH score, which consists of known coronary artery disease before OHCA, glucose ≥200 mg/dL, rhythm of arrest not ventricular tachycardia/fibrillation, age >45, and arterial pH ≤7.0 [7]. Lastly, the CAHP score model, which can be calculated using a previously published nomogram, consists of variables independently associated with poor neurologic outcome: age, arrest setting (location of arrest, public vs. home), initial rhythm (shockable or not), duration from the initial collapse to basic life support (BLS) and from BLS to ROSC, pH, and epinephrine dose (0, 1–2, 3) [6].

All of these models lack results from initial neurologic examinations (i.e., the Full Outline of UnResponsiveness brainstem reflexes (FOUR_B) score and Glasgow Coma Scale motor (GCS_M) score), which are simple examinations, have already been described as possible prognostic factors, and are mandatory work-ups for unconsciousness, but provide important information in predicting neurologic outcomes [8–10].

Recently, each prediction model has been compared with others, improved by adding new factors, or validated based on several other demographic features to overcome worldwide variations [1, 11–13]. However, there is still a shortage of external validation studies, and no

definitely established prediction model exists. Hence, the aim of this study is not only to confirm whether the existing models are applicable to the Korean population, but also whether prognostic accuracy increases when the cardiac arrest score is added to the initial neurologic examination.

## Materials and methods

### Setting

This study was performed in Seoul St. Mary's Hospital, Catholic University of Korea, which was established in August 1977 and is now designated as a tertiary care medical center with 1,356 beds, including 139 ICU beds. Approximately 70,000 patients visit its emergency centers every year, and OHCA patients are routinely treated with TTM in accordance with guideline recommendations [14, 15] based on usual cerebral performance category (CPC) scores. Briefly, TTM at 33˚C was induced using an endovascular cooling device (Thermoguard, ZOLL Medical Corporation, Chelmsford, MA, USA) or ArticSun (Bard Medical, Louisville, CO, USA) and maintained for at least 24 h. After the maintenance phase, rewarming to 37˚C was performed at a rate of 0.25˚C/h. Sedatives and neuromuscular blocking agents were routinely administered to control shivering before the induction of TTM. Percutaneous coronary intervention was available around the clock with a fast track protocol.

### Ethics

Our study protocol upheld the ethical guidelines of the 1975 Declaration of Helsinki and was approved by the Institutional Review Board of St. Mary's Hospital. Informed consent was not required due to the retrospective nature of this study. We completely anonymized all data before accessing them. We accessed all samples between 2009 and 2017. All samples were sourced from Seoul St. Mary's Hospital.

### Study population, data collection and outcome measure

A total of 311 OHCA patients treated with TTM in the intensive care unit between 2009 and 2017 were included in this study.

The laboratory data (levels of lactate, creatinine, glucose, pH, bicarbonate, sodium, potassium, etc.) were collected, and results of the initial neurologic examinations (FOUR_B and GCS_M), performed immediately after ROSC and before applying TTM, sedative, and muscle relaxant on the first day of admission, were obtained from the clinical record. Resuscitation data (initial rhythm, location of arrest, no-flow (time from collapse to initiation of CPR)/low-flow intervals (time from CPR to ROSC), injected epinephrine dose and times, and whether bystander CPR was conducted) were obtained from the situation reports of paramedics and clinical CPR records in the hospitals (if CPR was continued or rearrest occurred after arrival at the hospital). Sociodemographic data, medical histories, and comorbidities were collected from existing medical records or family/relative interviews.

We calculated three scoring models (OHCA, C-GRApH, and CAHP) related to survival with good neurologic recovery in accordance with the original publications [5–7]. In addition, results from the initial neurologic examinations (FOUR_B and GCS_M), which were possible prognostic factors, were added to the existing models with the expectation of a better prognosis. The primary outcome was neurologic outcome at the time of hospital discharge according to the 5 levels of the CPC scale (1–2 as good, 3–5 as poor).

## Statistical analysis

Normality tests were performed for continuous variables, and continuous variables are presented as means with standard deviations or as median values with interquartile ranges, as appropriate. Categorical variables are presented as frequencies and percentages. For patient characteristics and comparisons between groups, we used Student's t-test and the Mann-Whitney U test for continuous variables and Fisher's exact test and the Chi-squared test for categorical variables.

We determined predictive performance using receiving operating characteristic (ROC) curves set up with logistic regression models to assess and compare the equality of each area under the curve (AUC) using the Delong test. First, we determined the AUC for the OHCA, C-GRApH, and CAHP scores using the ROC curves (i.e., the area under the receiver operating characteristic curve (AUROC)). Then, to test the superiority of the addition of the FOUR_B and GCS_M scores compared with the OHCA, C-GRApH, and CAHP scores alone, the AUCs in combination with the FOUR_B and GCS_M scores were calculated and compared to those with the OHCA, C-GRApH, and CAHP scores. Statistical analyses were performed using SPSS 21.0 (Chicago, IL, USA) and MedCalc 15.2.2 (MedCalc Software, Mariakerke, Belgium). P values $\leq 0.05$ were considered statistically significant. The Youden Index was used to determine the optimal cutoff point for poor neurologic outcome.

## Results

### Baseline demographic characteristics

A total of 311 surviving CA patients treated with TTM between 2009 and 2017 were recruited for this retrospective study. Of these, 99 (31.8%) patients had a good neurologic outcome, and 212 patients had a poor neurologic outcome.

Baseline characteristics for patients stratified according to the CPC definition (CPC 1 and 2: good; CPC 3–5: poor) are shown in Table 1. Patients in the poor neurologic outcome group were older (median 48 vs. 58 years, $p < 0.001$) and had a higher presence of diabetes (27.8% vs. 9.1%, $p < 0.001$). However, the initial glucose levels did not differ between the two groups in this study. Resuscitation-related factors, such as arrest outside of the home, witnessed arrest, initial shockable rhythm (i.e., ventricular tachycardia and ventricular fibrillation), cardiac causation and short anoxic time (no flow and low flow) were more likely to be followed by a good neurologic outcome. Lactate and creatinine levels were higher in the poor outcome group, and the serum pH level was lower in the poor outcome group.

### External validation of existing prognostic scoring models and newly identified neurologic outcome-related factors

First, two groups were classified as good and poor according to the neurologic prognosis by applying the existing scoring models (OHCA, C-GRApH, and CAHP) following the guidelines of the original publications. The OHCA, C-GRApH and CAHP scores were 22.9 (95% confidence interval (CI): 11.5–34.3), 2.0 (95% CI: 1.0–2.0), and 129 (95% CI: 92.5–160), respectively, in the good neurologic outcome group and 46.1 (95% CI: 35.9–56.1), 3.0 (95% CI: 2.0–4.0) and 206 (95% CI: 171.3–243.8), respectively, in the poor neurologic outcome group (Table 2). The AUROC value of each model was found to be 0.844 (95% CI: 0.798–0.884), 0.779 (95% CI: 0.728–0.824) and 0.872 (95% CI: 0.830–0.907), respectively, in our cohort (Table 3). Table 4 shows the sensitivity, specificity, positive predictive value, and negative predictive value of each single test. The cutoff values of the OHCA, C-GRApH and CAHP scores were 30.91 (sensitivity

**Table 1. Demographic characteristics of subjects according to neurologic outcome at hospital discharge.**

|  | Good N = 99 | Poor N = 212 | p |
|---|---|---|---|
| Age, median (IQR) | 48 (39–61) | 58 (46–72) | < 0.001 |
| Sex, male | 75 (75.8) | 145 (68.4) | 0.228 |
| Past History |  |  |  |
| HTN | 21 (21.2) | 74 (34.9) | 0.017 |
| DM | 9 (9.1) | 59 (27.8) | < 0.001 |
| CAD | 15 (15.2) | 22 (10.4) | 0.260 |
| Arrest setting, home (%) | 37 (37.4) | 127 (59.9) | < 0.001 |
| Witnessed arrest, No. (%) | 83 (83.8) | 132 (62.3) | < 0.001 |
| Bystander CPR, No. (%) | 61 (61.6) | 115 (54.2) | 0.269 |
| Shockable rhythm, No. (%) | 71 (71.7) | 43 (20.3) | < 0.001 |
| Cardiac cause of arrest, No. (%) | 90 (90.9) | 113 (53.3) | < 0.001 |
| Anoxic time |  |  |  |
| No flow, min, median (IQR) | 3.5 (0–6.0) | 5.0 (1–12.3) | 0.017 |
| Low flow, min, median (IQR) | 14.5 (9.0–23.0) | 28.0 (20.0–37.3) | < 0.001 |
| Lactate, mmol/L | 6.6 (3.5–11.5) | 11.4 (7.0–14.9) | < 0.001 |
| Creatinine, mg/dl | 1.09 (0.94–1.30) | 1.30 (1.03–1.8) | < 0.001 |
| Glucose, mg/dl | 236.5 (185.3–285.5) | 281.5 (196.5–348.5) | 0.004 |
| pH | 7.25 (7.13–7.33) | 6.99 (6.85–7.18) | < 0.001 |
| Epinephrine (%) |  |  | < 0.001 |
| 0mg | 50 (56.8) | 12 (6.9) |  |
| 1–2 mg | 16 (18.2) | 53 (30.5) |  |
| ≥ 3 mg | 22 (25.0) | 109 (62.6) |  |

Variables are expressed as median (interquartile range) or n (%).

IQR, interquartile range; HTN, hypertension; DM, diabetes mellitus; CAD, coronary artery disease; CPR, cardiopulmonary resuscitation.

84.6%, specificity 70.8%), 2 (sensitivity 68.6%, specificity 76.8%), and 167 (sensitivity 78.3%, specificity 81.8%), respectively, for predicting poor neurologic outcome.

In the univariate analysis, the FOUR_B and GCS_M scores were also identified as significant indices of neurologic outcome-related factors (both P<0.001). To validate these factors,

**Table 2. Neurologic examination and prediction scores for predicting poor neurologic outcome at hospital discharge.**

|  | Good N = 99 | Poor N = 212 | p |
|---|---|---|---|
| OHCA score | 22.9 (11.5–34.3) | 46.1 (35.9–56.1) | < 0.001 |
| C-GRApH score | 2.0 (1.0–2.0) | 3.0 (2.0–4.0) | < 0.001 |
| CAHP score | 129 (92.5–160) | 206 (171.3–243.8) | < 0.001 |
| FOUR_B |  |  | < 0.001 |
| FOUR_B = 0,1 | 24 (24.7) | 170 (81.0) |  |
| FOUR_B = 2 | 19 (19.6) | 23 (11.0) |  |
| FOUR_B = 4 | 54 (55.7) | 17 (8.1) |  |
| GCS_M |  |  | < 0.001 |
| GCS_M = 1 | 44 (44.4) | 196 (93.3) |  |
| GCS_M>1 | 55 (55.6) | 14 (6.7) |  |

FOUR_B, Full Outline of UnResponsiveness Brainstem reflexes; GCS_M, Glasgow Coma Scale motor score.

**Table 3. AUROC values for neurologic examination and prediction scores to predict poor neurologic outcome at hospital discharge.**

|  | AUROC (95% CI) |
| --- | --- |
| FOUR_B | 0.804 (0.755–0.847) |
| GCS_M | 0.744 (0.692–0.792) |
| OHCA score | 0.844 (0.798–0.884) |
| C-GRApH score | 0.779 (0.728–0.824) |
| CAHP score | 0.881 (0.841–0.922) |

AUROC, area under receiver operating characteristic curve; CI, confidence interval; FOUR_B, Full Outline of UnResponsiveness Brainstem reflexes; GCS_M, Glasgow Coma Scale motor score.

patients were classified into 3 subgroups based on the FOUR_B score and 2 subgroups based on the GCS_M score. Based on the FOUR_B scores, with 4 candidates (2 each in the good and poor groups) excluded due to a lack of neurologic examination records, the remaining 307 patients of the 311 surviving patients were categorized into FOUR_B = 0–1, FOUR_B = 2 and FOUR_B = 4 subgroups. Based on the GCS_M scores, with 2 candidates with poor neurologic outcomes excluded due to a lack of records, the patients underwent a subgroup analysis by GCS_M = 1 and GCS_M > 1 (all P<0.001). As a result of applying these factors to the AUROCs, values of 0.804 (95% CI: 0.755–0.847) and 0.744 (95% CI: 0.692–0.792) were derived from the FOUR_B and GCS_M models, respectively, as shown in Tables 2 and 3.

## Enhanced prediction accuracy by adding FOUR_B and/or GCS_M scores to the existing scoring models

To verify that the prognostic predictive power increased by adding FOUR_B and GCS_M scores to the existing models, we used the AUROC values. Compared with using the OHCA score alone (AUROC: 0.844; 95% CI: 0.798–0.884), the OHCA score with FOUR_B (AUROC: 0.899; 95% CI: 0.858–0.931), OHCA score with GCS_M (AUROC: 0.880; 95% CI: 0.837–0.915), and OHCA score with FOUR_B and GCS_M (AUROC: 0.911; 95% CI: 0.873–0.941) models showed gradual improvements. The AUROC values were not significantly different for the C-GRApH score (AUROC: 0.779; 95% CI: 0.728–0.824), CAHP score (AUROC: 0.872,

**Table 4. Sensitivity, specificity, PPV, NPV for predicting neurologic outcome at hospital discharge.**

|  | Sensitivity (95% CI) | Specificity (95% CI) | PPV (95% CI) | NPV (95% CI) |
| --- | --- | --- | --- | --- |
| FOUR_B | 81.0 | 75.3 | 87.6 | 64.6 |
| > 1 | (75.0–86.0) | (65.5–83.5) | (82.2–91.9) | (55.0–73.4) |
| GCS_M | 93.3 | 55.6 | 81.7 | 79.7 |
| ≤ 1 | (89.1–96.3) | (45.2–65.5) | (76.2–86.4) | (68.2–88.5) |
| OHCA score | 84.6 | 70.8 | 85.9 | 68.7 |
| > 30.91 (Youden) | (78.8–89.3) | (60.7–79.7) | (80.2–90.4) | (58.5–77.7) |
| C-GRApH score | 68.6 | 76.8 | 86.2 | 53.5 |
| > 2 (Youden) | (61.8–74.8) | (67.2–84.7) | (80.1–91.1) | (44.9–62.0) |
| CAHP score | 78.3 | 81.8 | 90.2 | 63.8 |
| > 167 | (72.1–83.7) | (72.8–88.9) | (85.0–94.1) | (54.8–72.1) |

Variables are expressed as median (interquartile range) or n (%).

PPV, positive predictive value; NPV, negative predictive value; CI, confidence interval; FOUR_B, Full Outline of UnResponsiveness Brainstem reflexes; GCS_M, Glasgow Coma Scale motor score.

95% CI: 0.830–0.907), C-GRApH score with FOUR_B (AUROC: 0.877; 95% CI: 0.835–0.912), CAHP score with FOUR_B (AUROC 0.901, 95% CI 0.862–0.932), C-GRApH score with GCS_M (AUROC: 0.869; 95% CI: 0.826–0.905), CAHP score with GCS_M (AUROC 0.897, 95% CI 0.858–0.929), C-GRApH score with FOUR_B and GCS_M (AUROC: 0.906; 95% CI: 0.868–0.936), or CAHP score with FOUR_B and GCS_M (AUROC 0.913, 95% CI 0.875–0.942) (Figs 1–3).

## Discussion

### Main findings

In this retrospective study, the OHCA, C-GRApH, and CAHP scores shown in the Korean population were all similar to those in the original publications by using AUROC. Additionally, newly identified neurologic outcome-related factors (FOUR_B and GCS_M scores) were found to be significant variables in neurologic outcome prediction. By adding these variables to the original models, the accuracy of the neurologic outcome prediction was further

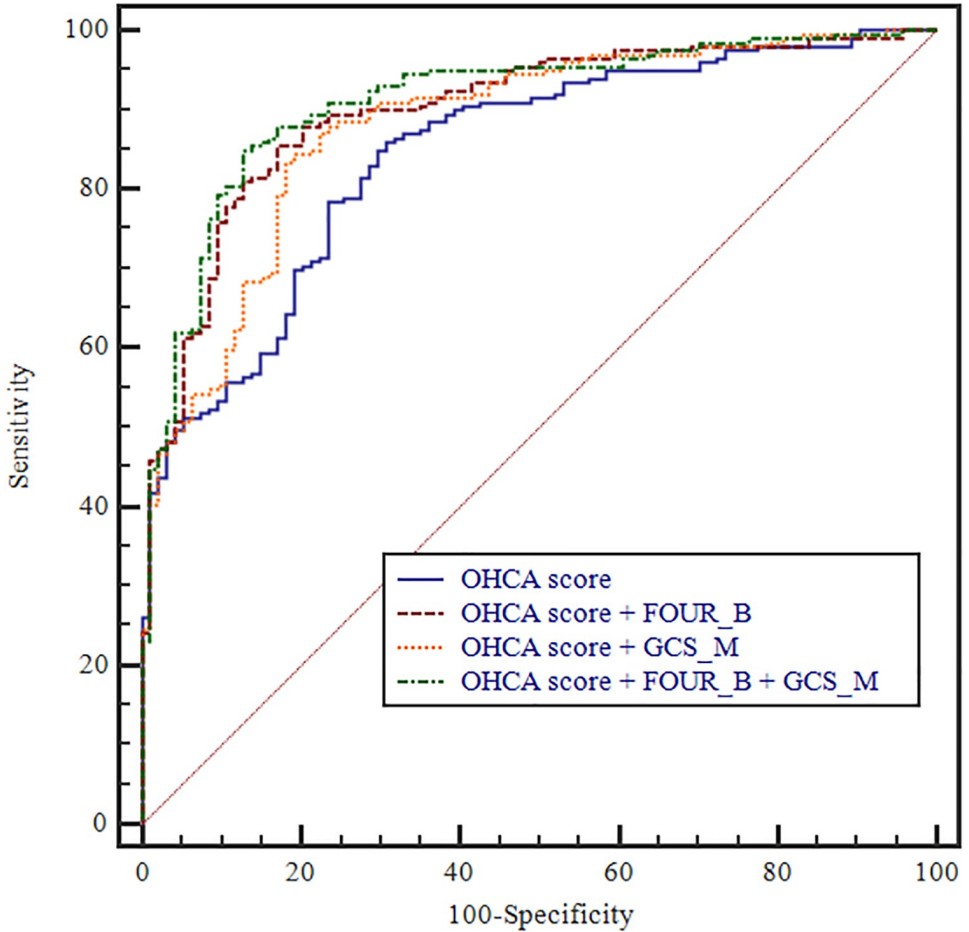

**Fig 1. Prognostic value of OHCA score for the prediction of poor neurologic outcome after CA.** [1] OHCA score (AUC 0.844, 95% CI 0.798–0.884); [2] OHCA score with FOUR_B (AUC 0.899, 95% CI 0.858–0.931); [3] OHCA score with GCS_M (AUC 0.880, 95% CI 0.837–0.915); [4] OHCA score with FOUR_B and GCS_M (AUC 0.911, 95% CI 0.873–0.941).

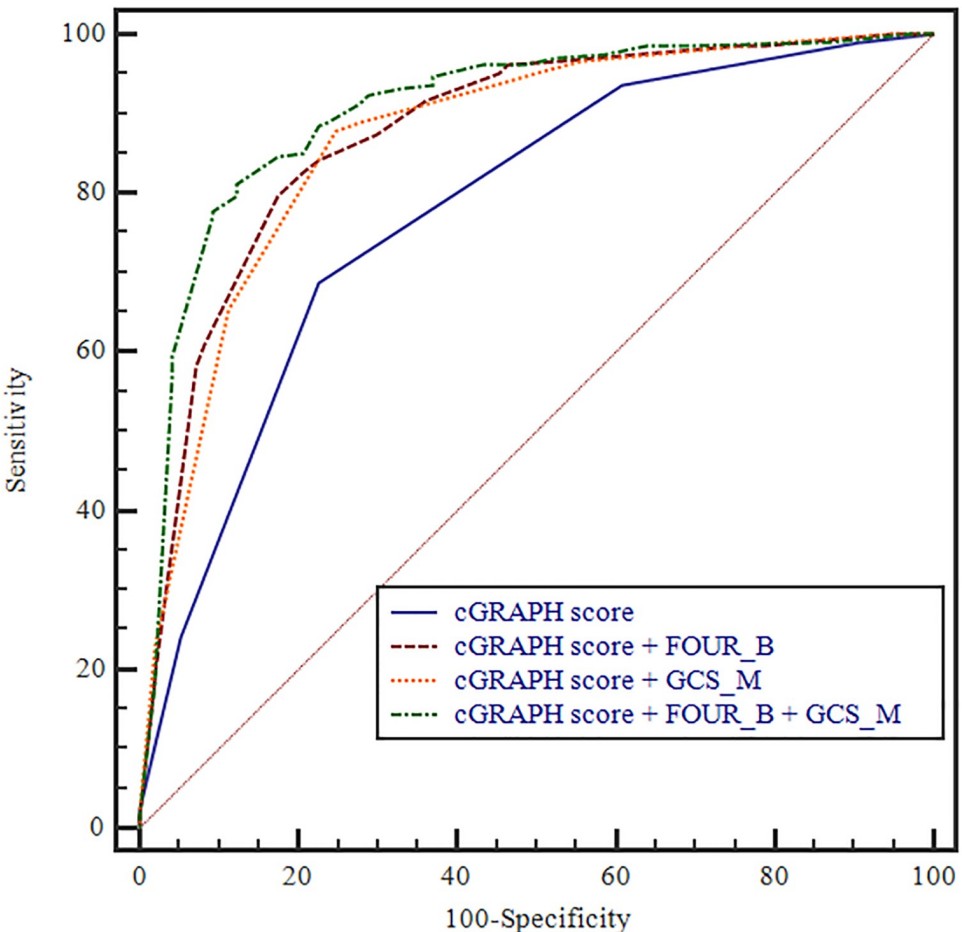

**Fig 2. Prognostic value of C-GRApH score for the prediction of poor neurologic outcome after CA.** [1] C-GRApH score (AUC 0.779, 95% CI 0.728–0.824); [2] C-GRApH score with FOUR_B (AUC 0.877, 95% CI 0.835–0.912); [3] C-GRApH score with GCS_M (AUC 0.869, 95% CI 0.826–0.905); [4] C-GRApH score with FOUR_B and GCS_M (AUC 0.906, 95% CI 0.868–0.936).

enhanced. Of those, the AUROC (0.913, 95% CI 0.875–0.942) was the highest when the CAHP score, FOUR_B and GCS_M were combined.

## Importance of the prediction of neurologic prognoses and limitations of the existing models

The neurologic prognosis of post-CA patients is a concern worldwide. Since WLST is limited in Korea, the random removal of tracheal intubation, vasopressors and/or inotropes), and continuous renal replacement therapy performed in patients with hypoxic brain damage after CA is prohibited by law. As a result, most patients are shifted to conservative treatment centers if recovery is considered to be difficult. Thus, early assessment of the neurologic prognosis of patients has important implications, not only for patients from the perspective of dignity of life, but also for their families and our society to reduce the burden of continued economic costs.

One of the fatal drawbacks of the OHCA score is anoxic time-related factors, such as no-flow and low-flow intervals. These are known to be the main hurdle, not only because they are

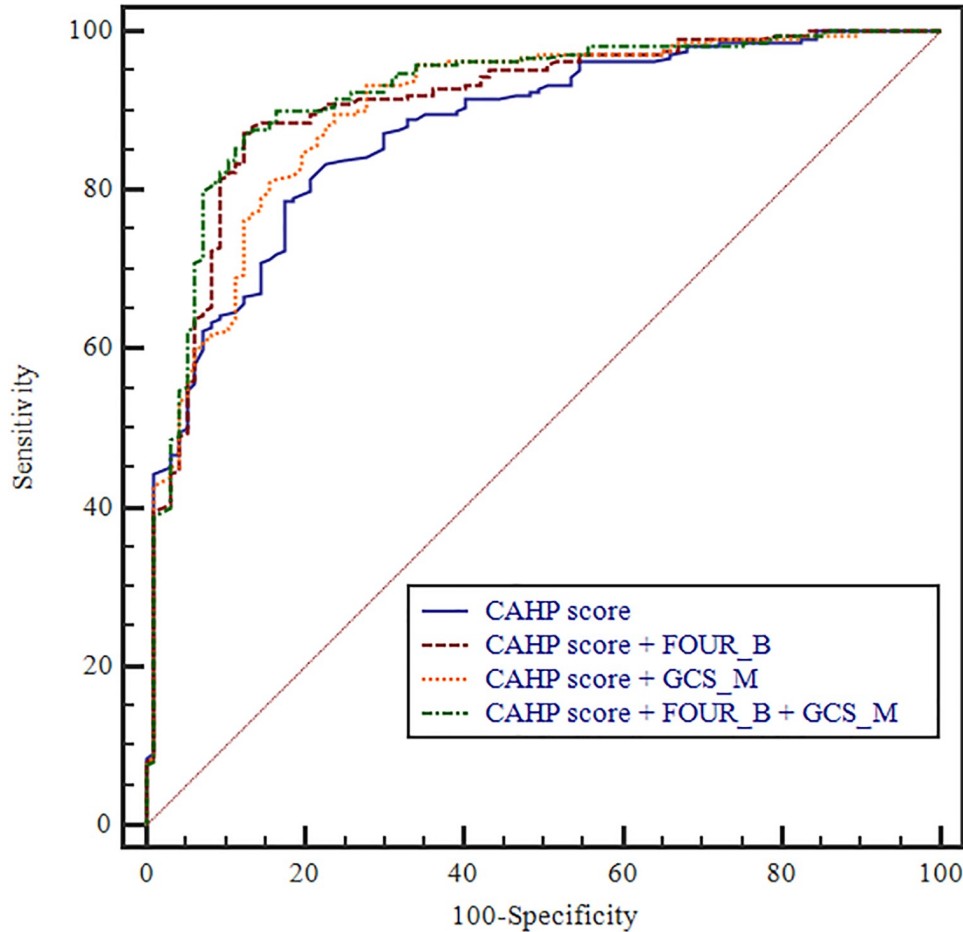

**Fig 3. Prognostic value of CAHP score for the prediction of poor neurologic outcome after CA.** [1] CAHP score (AUC 0.872, 95% CI 0.830–0.907); [2] CAHP score with FOUR_B (AUC 0.901, 95% CI 0.862–0.932); [3] CAHP score with GCS_M (AUC 0.897, 95% CI 0.858–0.929); [4] CAHP score with FOUR_B and GCS_M (AUC 0.913, 95% CI 0.875–0.942).

difficult to calculate without witnesses (if a witness does not exist, the estimates are categorical), but also because an additional single minute could aggravate the prognosis by 7–10% even though a logarithmic scale is applied [16]. Furthermore, the accuracy related to estimates of resuscitation delays is CAHP's pre-acknowledged limitation [6]. Despite this limitation, many researchers do not exclude time-related factors even if they accept the limitations, and reducing this time is the basis of CPR public education [6, 17]. In the results of this study, as shown in Table 1, all of the anoxic time-related factors (no flow and low flow) showed statistical significance with a p value < 0.001. Among the variables included in the CAHP score, the patients were older and arrest at home was more common in the poor neurologic outcome group. According to BK Lee et al., age was not associated with in-hospital mortality in a setting where WLST was not performed, but further research on this is needed [18]. It also remains unclear whether the criteria regarding glucose levels should be regarded as simply the presence/absence of diabetes mellitus or should be classified based on specific glucose levels [19].

Although each prediction model has some limitations, we applied these models to our single-centered cohort, and the AUROC-based prediction values were in line with previous validation studies, as shown in Table 3.

## Newly identified factors related to the initial neurologic test and its application to the existing model

To improve the accuracy of neurologic outcome prediction and to account for each of the limitations, we can add several other important factors related to neurologic evaluations and reduce or ultimately control the weight of each factor.

First, the GCS score was selected as one of the main predictive factors. Although it consists of 3 factors (eye, verbal and motor responses), the motor response was selected because the other two elements cannot be distinguished in CA patients who are usually intubated in a coma state [20]. Second, the brainstem response of the FOUR score was assessed. According to Wijdicks and colleagues, the FOUR score consists of four components (eye, motor, brainstem, and respiration), and each component has a maximal score of 4 [21]. During CA, eye, motor, and respiration cannot be fully assessed or show no differences; however, brainstem reflexes that control basic functions such as breathing, swallowing, heart rate, blood pressure, consciousness, and whether one is awake or asleep can be somewhat distinguishable by B0, 1, 2 and 4 (if B3 is observed, the patient is not an appropriate candidate for TTM).

Both neurologic grading scales have been used to assess patients with impaired levels of consciousness and as mandatory exams for mental changes in patients, and these tests can be performed in a very short amount of time with a simple bedside test. Although the GCS_M score and FOUR score tend to overlap, due to the limitations in normal responses in CA patients, these grading scales are not identical. While the three prognostic prediction models presented above can predict the neurologic prognosis using indirect factors resulting from hypoxia damage-related metabolic changes in the body, recent studies have tended to estimate the neurologic outcome using brain imaging modalities such as computed tomography (CT) and magnetic resonance imaging, which are directly related to hypoxic injury in the brain. Early brain damage is evaluated by the gray matter to white matter attenuation ratio on head CT, which can distinguish the severity of brain edema [8]. However, consensus has not yet been established on the appropriate time for CT to be taken, for example, immediately after ROSC, to reflect an initial injury of the brain [22–24]. Recent trends related to imaging modalities have suggested that factors related to neurologic tests should be included to enhance prediction accuracy, as the existing models lack important information directly related to brain damage.

The significance of GCS has already been demonstrated in pediatric patients, and articles on the association between the FOUR score and prognostic prediction have recently been published [8–10]. In this study, newly identified factors related to the initial neurologic test, i.e., FOUR_B and GCS_M scores, were added to OHCA, C-GRApH, and CAHP scores that were externally validated using AUROC. With the OHCA score, the AUROC significantly increased from 0.844 (95% CI: 0.798–0.884) to 0.911 (95% CI: 0.873–0.941). The results were similarly increased from 0.779 (95% CI: 0.728–0.824) to 0.906 (95% CI: 0.868–0.936) for the C-GRApH score and from 0.872 (95% CI 0.830–0.907) to 0.913 (95% CI 0.875–0.942) for the CAHP score.

## Clinical meaning of enhanced prediction by adding initial neurologic test on existing scoring models and its limitation

Prognostication after cardiac arrest is challenging. Although several tools (such as pupillary and corneal reflexes, absence of the N20 wave of somatosensory evoked potentials, concentration of neuron-specific enolase, electroencephalography and brain diffusion weighted imaging) have been studied, no tool can predict the patient's neurologic outcome with 100% certainty [25, 26]. Moreover, TTM influences the metabolism of sedative drugs and may interfere with an accurate

prognostication [27]. Nevertheless, accurate prediction after cardiac arrest is essentially needed to minimize the risk of erroneous prognostication of poor outcome.

It is true that the results of this research (AUROC 0.8–0.9) should not be used in WLST, which is the leading cause of death after CA [28]. In terms of outcome prediction, early prediction, as well as accurate prediction, is very important. It is possible to allocate critical care resources (early coronary angiogram, extracorporeal membrane oxygenation, etc.) by predicting the patient prognosis early. The outcome prediction tools presented in the guidelines have a high prediction accuracy, but limitations regarding the time–a delay of 3 to 7 days after TTM depending on each protocol–are also obvious. WLST after ROSC in Korea is legally prohibited, and early outcome prediction is crucial in regard to the proper distribution of limited medical assets.

This study has several limitations. First, it is a small-sample, retrospective study conducted at a single institution. Therefore, additional external validation should be performed in various contexts. Second, even though we knew the problems associated with the existing models, an immediate solution was not identified. However, we tried to attenuate the problem by identifying and including new factors, and it is meaningful that this article showed the possibility that accuracy can be increased by including additional significant factors.

## Conclusion

The OHCA, C-GRApH and CAHP scores all had good prognostic performance for predicting the neurologic outcome in OHCA patients treated with TTM. By adding new variables associated with the initial neurologic examinations, the prognoses of neurologic outcomes improved compared to the existing scoring models. Of those, the AUROC (0.913, 95% CI 0.875–0.942) was the highest when the CAHP score, FOUR_B and GCS_M were combined. Large, multi-center studies should verify a definite prognostic model in the near future by allocating the appropriate weight for each factor.

## Supporting information

**S1 File.**
(XLSX)

## Author Contributions

**Conceptualization:** Kyu Nam Park, Soo Hyun Kim, Sang Hoon Oh, Chun Song Youn.

**Data curation:** Chun Song Youn.

**Formal analysis:** Hyun Soo Kim.

**Investigation:** Chun Song Youn.

**Methodology:** Kyu Nam Park, Soo Hyun Kim, Byung Kook Lee, Kyung Woon Jeung, Seung Pill Choi, Chun Song Youn.

**Writing – original draft:** Hyun Soo Kim.

**Writing – review & editing:** Kyung Woon Jeung, Chun Song Youn.

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
