## [Decision Letter · Decision Letter 0]

17 Feb 2020

PONE-D-20-01272

Prognostic value of OHCA and C-GRApH scores to predict neurologic outcomes in cardiac arrest patients treated with targeted temperature management

PLOS ONE

Dear Dr. Youn,

Thank you for submitting your manuscript to PLOS ONE. After careful consideration, we feel that it has merit but does not fully meet PLOS ONE’s publication criteria as it currently stands. Therefore, we invite you to submit a revised version of the manuscript that addresses the points raised during the review process.

We would appreciate receiving your revised manuscript by Apr 02 2020 11:59PM. To enhance the reproducibility of your results, we recommend that if applicable you deposit your laboratory protocols in protocols.io, where a protocol can be assigned its own identifier (DOI) such that it can be cited independently in the future. For instructions see: http://journals.plos.org/plosone/s/submission-guidelines#loc-laboratory-protocols

We look forward to receiving your revised manuscript.

Kind regards,

Corstiaan den Uil

Academic Editor

PLOS ONE

Journal Requirements:

2. In the ethics statement in the manuscript and in the online submission form, please provide additional information about the patient records used in your retrospective study, including: a) whether all data were fully anonymized before you accessed them and b) the date range (month and year) during which patients' medical records were accessed.

"This research was supported by Basic Science Research Program through the National Research Foundation of Korea (NRF) funded by the Ministry of Education (2018R1D1A1B07047594)"

"The authors received no specific funding for this work."

Reviewers' comments:

Reviewer's Responses to Questions

**Comments to the Author**

1. Is the manuscript technically sound, and do the data support the conclusions?

Reviewer #1: Yes

Reviewer #2: Partly

2. Has the statistical analysis been performed appropriately and rigorously? 

Reviewer #1: Yes

Reviewer #2: Yes

3. Have the authors made all data underlying the findings in their manuscript fully available?

Reviewer #1: Yes

Reviewer #2: Yes

4. Is the manuscript presented in an intelligible fashion and written in standard English?

Reviewer #1: Yes

Reviewer #2: Yes

5. Review Comments to the Author

Reviewer #1: Thank you for this interesting manuscript which is an enjoyable reading and might be interesting for the readers of PlosOne.

The manuscript submitted by Youn et al. aims to the prognostic value of OHCA and C-GRApH scores for predicting neurologic outcomes. The authors report results of a retrospective validation of OHCA and C-GRApH score for neurologic outcome at 6 months after CA. And found added FOUR_B or GCS_M scores to the OHCA score and C-GRApH score improved the prediction of poor neurologic outcome (0.84 -> 0.89, 0.77 -> 0.88).

Major comments:

1. All in all, the analysis is well done, the results are comprehensible and offer a good statement, because the analysis appears very clear. However, Fundamentally I have concern about why we need a scoring system that predicting poor prognosis with AUC of 0.8~0.9. Current guidelines emphasize the need to wait for some time after the return to normothermia to assess the likelihood of poor neurological outcome and minimize false-positive rates (less than 1%). Further discussion will be needed for the clinical application of the authors’ scoring system for decision making in early OHCA survivors.

2. Study population

Authors include 311 adult patients during 8 years without excluding (trauma?). I wonder how many OHCA patients and get ROSC during study period? This is important information to validate scoring system to reduced selection bias.

3. 6-month CPC score

It is very impressive collect 6-month CPC score all included patients, even though it is retrospective study design. It’s interesting to state how to get a 6-month CPC score in Method section.

Minor

- check the abbreviation again: ex. ROSC in Introduction, AUROC in Method, PPV,NPV in Results, WLST, CRRT, DM etc.

- booster in Discussion means vasopressor?

Reviewer #2: Dear Authors, Thanks for your manuscript full of facts. It was interesting to read. The relevance of specific scores after OHCA is an important issue. Clinical examination on admission is also important and to merge these two aspects seems to me original and relevant.

Nevertheless, I think the manuscript could be enhanced by adressing the following comments :

Major comments :

1/ It seems to me that the interesting part of your work is the added value of clinical examination with the scores. Your work should not be presented as a validation of these scores (most already validated elsewhere) but as what had the clinical examination to these scores (Title for example)

2/ You choose to investigate the pronostic value of OHCA and C-GRApH scores. Why did you choose these scores ? OHCA score is an historical but « old » score and has been already externaly validated in many studies in European and US cohorts. C-GRApH score seems to me more « confidential ». A recent other french developed score, the CAHP score, seems to me important and is described in many recent publications.

I suggest that you had the CAHP score at your work. I also suggest that you had a statement to explain why you choose these scores.

3/ It should be clearly stated when were the patients clinically tested. If this work is in the setting of TTM, at which T° where tested the patients ? I understand that it is on day 1, probably on arrival but precisely ? Were the patients under sedation ?...

Minor comments :

Intro :

-In OHCA score the possible errors due to NF and LF estimation is reduced by the logarithmic transformation. Disadvantage is : you can’t compute the score if you do not have a NF (i.e no whitness).

-Aim of the study : should be rewritten (see Major 1/)…. « to test the added value of initial clinical examination, on prognostic scores … »

Math Meth

Setting :

-In this paragraph, you should explain what is your TTM. When and How are your patients cooled ? Which target ? (see Major 3/)

Study population :

-Only state state that « adult OHCA patients admitted to the ICU were included ». (The sentence « There were no exclusion criteria except… » is to complicated)

-Primary outcome : you write that primary outcome was neurologic outcome at 6 months. However all the presented scores tested neurological outcome at hospital discharge. Please align that in the paper.

Results :

-Delete « after eliminating…..due to age requirements, » Only state : « Three hundred and one OHCA adult patients have been included in the study »

-« Old age showed poor neurologic outcome » : it is not exactly what your data show ; you only show that patients with poor neurological outcome are older…

-About Relative and Absolute anoxic time, see under « Table 1 « comments.

-Total anoxic time is not very relevant in the OHCA literature. It could be deleted.

-High lactate level and low pH are NEVER associated with satisfactory neurological outcome… Please correct the sentence.

-« Enhanced prediction accuracy… »

I don’t understant what means the first paragraph of this section from « as mentioned above..to … might be further enhanced ». Delete ?

Discussion :

Main findings : ok

The importance of the prediction.. :

-Why not explain in the introduction the problem of WLST in Korea ? It justifies your work and give light on why you perform it.

-The discussion on age should be rewritten (see above about the misinterpretation of the age in the results)

Conclusion : Agree with.

What is your proposition of the best score combined with cliniacl examination ?

Table 1 :

-Please write clearly what is HTN, DM and CAD…

-Why using Relative and Absolute anoxic time ? Is it NF and LF ? It differs from the Materiel and Methods section. Please align.

6. PLOS authors have the option to publish the peer review history of their article (what does this mean?). If published, this will include your full peer review and any attached files.

Reviewer #1: No

Reviewer #2: Yes: Bertrand Sauneuf

---

## [Author Response · Author response to Decision Letter 0]

31 Mar 2020

Reply to reviewer

Thank you for your kind consideration. I did my best to review the contents as you had recommended. The manuscript has much improved according to reviewer’s comments.

Reviewer #1: Thank you for this interesting manuscript which is an enjoyable reading and might be interesting for the readers of PlosOne.

The manuscript submitted by Youn et al. aims to the prognostic value of OHCA and C-GRApH scores for predicting neurologic outcomes. The authors report results of a retrospective validation of OHCA and C-GRApH score for neurologic outcome at 6 months after CA. And found added FOUR_B or GCS_M scores to the OHCA score and C-GRApH score improved the prediction of poor neurologic outcome (0.84 -> 0.89, 0.77 -> 0.88).

Major comments:

1. All in all, the analysis is well done, the results are comprehensible and offer a good statement, because the analysis appears very clear. However, Fundamentally I have concern about why we need a scoring system that predicting poor prognosis with AUC of 0.8~0.9. Current guidelines emphasize the need to wait for some time after the return to normothermia to assess the likelihood of poor neurological outcome and minimize false-positive rates (less than 1%). Further discussion will be needed for the clinical application of the authors' scoring system for decision making in early OHCA survivors.

Thank you for pointing out the important part. I absolutely agree with your comments. As pointed out, it is true that the results of this research (AUC 0.8-0.9) is insufficient compared to the methods of MRI, EP, EEG suggested by guideline after therapeutic hypothermia. However, as you already know, for patients with post cardiac arrest prognosis prediction is very important part, and early prognosis prediction as well as accuracy are needed. The existing method has high prediction accuracy, but the limitations that are delayed by the method that can be implemented 3-5days after therapeutic hypothermia is implemented are clear. In addition, especially in Korea’s medical environment, withdrawal after ROSC is not only impossible due to ethical problem, but early prognosis prediction will be a very important part in proper distribution of limited medical assets. We added following to the discussion.

Prognostication after cardiac arrest is challenging. Although several tools (such as pupillary and corneal reflexes, absence of the N20 wave of somatosensory evoked potentials, concentration of neuron-specific enolase, electroencephalography and brain diffusion weighted imaging) have been studied, no tool can predict the patient's neurologic outcome with 100% certainty (25, 26). Moreover, TTM influences the metabolism of sedative drugs and may interfere with an accurate prognostication (27). Nevertheless, accurate prediction after cardiac arrest is essentially needed to minimize the risk of erroneous prognostication of poor outcome.

It is true that the results of this research (AUROC 0.8-0.9) should not be used in WLST, which is the leading cause of death after CA (28). In terms of outcome prediction, early prediction, as well as accurate prediction, is very important. It is possible to allocate critical care resources (early coronary angiogram, extracorporeal membrane oxygenation, etc.) by predicting the patient prognosis early. The outcome prediction tools presented in the guidelines have a high prediction accuracy, but limitations regarding the time – a delay of 3 to 7 days after TTM depending on each protocol – are also obvious. WLST after ROSC in Korea is legally prohibited, and early outcome prediction is crucial in regard to the proper distribution of limited medical assets.

2. Study population

Authors include 311 adult patients during 8 years without excluding (trauma?). I wonder how many OHCA patients and get ROSC during study period? This is important information to validate scoring system to reduced selection bias.

Thank you for your comments. I absolutely agree with your comment. Followings are the exclusion criteria for TTM after cardiac arrest in our hospital which was published before.

All unconscious adult patients who were resuscitated from nontraumatic OHCA and brought to the emergency department were eligible for the TTM. Patients were excluded from the TTM according to the following criteria: evidence of significant active bleeding, any intracranial hemorrhage, hemodynamic instability unresponsive to volume resuscitation and vasopressor treatment, severe dysrhythmia unresponsive to antiarrhythmic therapy, known terminal illness, poor prearrest neurologic status, and those with a ‘‘do not attempt resuscitation’’ preference. Ultimately, deciding whether a TH was initiated was up to the discretion of the treating physicians.

[REF] Youn CS, Kim SH, Oh SH, Kim YM, Kim HJ, Park KN. Successful implementation of comprehensive packages of postcardiac arrest care after Out-of-hospital cardiac arrest: a single institution experience in South Korea. Ther Hypothermia Temp Manag 2013;3:17–23.

The following table shows number of patients with OHCA, ROSC and treated with TTM during study period.

 2009 – 2017 

(study period of current study) 2009.1 – 2010.09

(study period of reference)

OHCA 1280 168

ROSC 572 (44.7%) 76 (45.2%)

TTM 311 (54.4%) 36 (47.4%)

The proportion of patients with ROSC is similar between two periods. However, the proportion of patients undergoing TTM treatment increased. This may be due to active implementation of ECMO early CAG, etc. Therefore, although this is retrospective study, the authors believe that there is little possibility of selection bias.

3. 6-month CPC score

It is very impressive collect 6-month CPC score all included patients, even though it is retrospective study design. It's interesting to state how to get a 6-month CPC score in Method section.

Thank you for your comments. Follow-up were performed by reviewing the electronic medical records or face-to-face interviews or telephone interviews by research specialist who is not involved in current study. Since 2015, our hospital has been conducting face-to-face studies to measure the level of cognitive function, depression, and fatigue in patients who have been discharged alive at 6 months after cardiac arrest. CPC scores at 6 months for patients before 2015 were obtained through telephone interviews or reviewing medical records. So, follow-up were performed by reviewing the electronic medical records or face to face interviews or telephone interviews. However, reviewer 2 indicated to change primary outcome to CPC on discharge in order to align with previous papers regarding OHCA score, cGRAPH score and CAHP score. So I change the primary outcome as CPC at discharge. Below is the change in CPC score at discharge and CPC score at 6 months

At discharge At 6 months

CPC 1 (n=90) CPC 1 (n=90)

CPC 2 (n=9) CPC 1 (n=4)

 CPC 2 (n=4)

 CPC 5 (n=1)

CPC 3 (n=13) CPC 2 (n=2)

 CPC 3 (n=7)

 CPC 4 (n=1)

 CPC 5 (n=3)

CPC 4 (n=60) CPC 3 (n=2)

 CPC 4 (n=19)

 CPC 5 (n=39)

CPC 5 (n=139) CPC 5 (n=139)

Minor

- check the abbreviation again: ex. ROSC in Introduction, AUROC in Method, PPV,NPV in Results, WLST, CRRT, DM etc.

Thank you for pointing out. I corrected. 

- booster in Discussion means vasopressor?

Thank you for pointing out. I deleted the term “booster’ in the manuscript.

Reply to reviewer

Thank you for your kind consideration. I did my best to review the contents as you had recommended. The manuscript has much improved according to reviewer’s comments.

Reviewer #2: Dear Authors, Thanks for your manuscript full of facts. It was interesting to read. The relevance of specific scores after OHCA is an important issue. Clinical examination on admission is also important and to merge these two aspects seems to me original and relevant.

Nevertheless, I think the manuscript could be enhanced by addressing the following comments:

Major comments :

1/ It seems to me that the interesting part of your work is the added value of clinical examination with the scores. Your work should not be presented as a validation of these scores (most already validated elsewhere) but as what had the clinical examination to these scores (Title for example)

Thank you for pointing it out. I absolutely agree with your comments. I accepted your opinion and changed the title to the following

Title; Prognostic value of OHCA, C-GRApH and CAHP scores with initial neurologic examinations to predict neurologic outcomes in cardiac arrest patients treated with targeted temperature management

2/ You choose to investigate the pronostic value of OHCA and C-GRApH scores. Why did you choose these scores ? OHCA score is an historical but « old » score and has been already externaly validated in many studies in European and US cohorts. C-GRApH score seems to me more « confidential ». A recent other french developed score, the CAHP score, seems to me important and is described in many recent publications.

I suggest that you had the CAHP score at your work. I also suggest that you had a statement to explain why you choose these scores.

Thanks for suggesting the CAHP score. Not only OHCA and C-GRApH mentioned in this study, but the CAHP score is a meaningful prognostic prediction methods. The purpose of this study was not to validate the cardiac arrest scores, but to verify that prognostic accuracy increases when cardiac arrest score is added to initial neurologic examination. In that sense, the reason for selecting OHCA and C-GRApH score is calculation convenience. CAHP score does not include initial neurologic examination. According to your comment, we added the CAHP score to the text. I corrected the aim of this study as following.

Hence, the aim of this study is not only to confirm whether the existing models are applicable to the Korean population, but also whether prognostic accuracy increases when the cardiac arrest score is added to the initial neurologic examination.

3/ It should be clearly stated when were the patients clinically tested. If this work is in the setting of TTM, at which T° where tested the patients ? I understand that it is on day 1, probably on arrival but precisely? Were the patients under sedation ?...

Thank you for your comments. I agree there were some confusing. 

According to our protocol, if a patient transferred by EMT with CPR became ROSC, non-enhanced brain CT is performed prior to applying therapeutic hypothermia to differentiated brain hemorrhage which is major contraindication of hypothermia therapy (exclusion criteria : a. acute hemorrhage evidence, b. usual activity score counting by CPC score(>3) and mRS(>3) before collapse.) The neurologic exam was performed immediately after the patient becomes ROSC, initial lab progressed, and subsequent labs were tested at 12hr after, 24hr after, and 48hr after based on the ROSC time. Therefore, all patients were tested immediately after ROSC and before applying TTM, sedative, and muscle relaxant. I added the following in the Method section.

The laboratory data (levels of lactate, creatinine, glucose, pH, bicarbonate, sodium, potassium, etc.) were collected, and results of the initial neurologic examinations (FOUR_B and GCS_M), performed immediately after ROSC and before applying TTM, sedative, and muscle relaxant on the first day of admission, were obtained from the clinical record.

Minor comments :

Intro :

-In OHCA score the possible errors due to NF and LF estimation is reduced by the logarithmic transformation. Disadvantage is : you can't compute the score if you do not have a NF (i.e no whitness).

Thank you for your comments. I deleted the following sentence in the Introduction section. 

However, the model was limited due to both the estimation difficulty of the no-flow and low-flow intervals, such that small errors resulted in very large differences in estimation, and the small sample sizes and selected cohorts.

-Aim of the study : should be rewritten (see Major 1/)…. « to test the added value of initial clinical examination, on prognostic scores … »

Thank you for your comments. I changed the aim of the study as follows.

Hence, the aim of this study is not only to confirm whether the existing models are applicable to the Korean population, but also whether prognostic accuracy increases when the cardiac arrest score is added to the initial neurologic examination.

Math Meth

Setting :

-In this paragraph, you should explain what is your TTM. When and How are your patients cooled ? Which target ? (see Major 3/)

Thank you for your comment. I added the following in the Method section.

. Briefly, TTM at 33°C was induced using an endovascular cooling device (Thermoguard, ZOLL Medical Corporation, Chelmsford, MA, USA) or ArticSun (Bard Medical, Louisville, CO, USA) and maintained for at least 24 h. After the maintenance phase, rewarming to 37°C was performed at a rate of 0.25°C/h. Sedatives and neuromuscular blocking agents were routinely administered to control shivering before the induction of TTM.

Study population :

-Only state state that « adult OHCA patients admitted to the ICU were included ». (The sentence « There were no exclusion criteria except… » is to complicated)

Thank you for your comment. I agree there was some confusion. I changed the sentence as follow.

A total of 311 OHCA patients treated with TTM in the intensive care unit between 2009 and 2017 were included in this study.

-Primary outcome : you write that primary outcome was neurologic outcome at 6 months. However all the presented scores tested neurological outcome at hospital discharge. Please align that in the paper.

Thank you for your comments. As you mentioned, the primary outcome was changed to a neurological outcome at hospital discharge to match the original publication. Below is the change in CPC score at discharge and CPC score at 6 months

At discharge At 6 months

CPC 1 (n=90) CPC 1 (n=90)

CPC 2 (n=9) CPC 1 (n=4)

 CPC 2 (n=4)

 CPC 5 (n=1)

CPC 3 (n=13) CPC 2 (n=2)

 CPC 3 (n=7)

 CPC 4 (n=1)

 CPC 5 (n=3)

CPC 4 (n=60) CPC 3 (n=2)

 CPC 4 (n=19)

 CPC 5 (n=39)

CPC 5 (n=139) CPC 5 (n=139)

Results :

-Delete « after eliminating…..due to age requirements, » Only state : « Three hundred and one OHCA adult patients have been included in the study »

Thank you for your comment. According to your comment, we changed the sentence as follow. 

A total of 311 surviving CA patients treated with TTM between 2009 and 2017 were recruited for this retrospective study.

-« Old age showed poor neurologic outcome » : it is not exactly what your data show ; you only show that patients with poor neurological outcome are older…

Thank you for your comment. According to your comment, we changed the sentence as follow. 

Patients in the poor neurologic outcome group were older (median 48 vs. 58 years, p < 0.001) and had a higher presence of diabetes (27.8% vs. 9.1%, p < 0.001).

-About Relative and Absolute anoxic time, see under « Table 1 « comments.

Thank you for your comment. I changed the term as no flow and low flow as you recommended.

-Total anoxic time is not very relevant in the OHCA literature. It could be deleted.

Thank you for your comment. I deleted the term “total anoxic time” as you recommended.

-High lactate level and low pH are NEVER associated with satisfactory neurological outcome… Please correct the sentence.

Thank you for your comment. According to your comment, we changed the sentence as follow.

Lactate and creatinine levels were higher in the poor outcome group, and the serum pH level was lower in the poor outcome group. 

-« Enhanced prediction accuracy… »

I don't understant what means the first paragraph of this section from « as mentioned above..to … might be further enhanced ». Delete ?

Thank you for your comments. I agree there were some confusing. I deleted the sentences as you recommended. 

Discussion :

Main findings : ok

The importance of the prediction.. :

-Why not explain in the introduction the problem of WLST in Korea ? It justifies your work and give light on why you perform it.

Thank you. Additionally mentioned to justified my work

-The discussion on age should be rewritten (see above about the misinterpretation of the age in the results)



Conclusion : Agree with.

What is your proposition of the best score combined with clinical examination ?

Thank you for your comment. I added the following sentence in the Conclusion section.

Of those, the AUROC (0.913, 95% CI 0.875–0.942) was the highest when the CAHP score, FOUR_B and GCS_M were combined.

Table 1 :

-Please write clearly what is HTN, DM and CAD…

Thank you for your comment. I edited as you recommended,

-Why using Relative and Absolute anoxic time ? Is it NF and LF ? It differs from the Materiel and Methods section. Please align.

 � Thank you for your comment. I changed the term as no flow and low flow as you recommended.

---

## [Editor Report · Decision Letter 1]

10 Apr 2020

Prognostic value of OHCA, C-GRApH and CAHP scores with initial neurologic examinations to predict neurologic outcomes in cardiac arrest patients treated with targeted temperature management

PONE-D-20-01272R1

Dear Dr. Youn,

We are pleased to inform you that your manuscript has been judged scientifically suitable for publication and will be formally accepted for publication once it complies with all outstanding technical requirements.

With kind regards,

Corstiaan den Uil

Academic Editor

PLOS ONE
---

## [Editor Report · Acceptance letter]

14 Apr 2020

PONE-D-20-01272R1 

Prognostic value of OHCA, C-GRApH and CAHP scores with initial neurologic examinations to predict neurologic outcomes in cardiac arrest patients treated with targeted temperature management 

Dear Dr. Youn:

I am pleased to inform you that your manuscript has been deemed suitable for publication in PLOS ONE. Congratulations! Your manuscript is now with our production department. 

With kind regards,

on behalf of

Dr. Corstiaan den Uil 

Academic Editor

PLOS ONE